# Development of a Novel Anti-CD44 Variant 8 Monoclonal Antibody C_44_Mab-94 against Gastric Carcinomas

**DOI:** 10.3390/antib12030045

**Published:** 2023-07-04

**Authors:** Hiroyuki Suzuki, Nohara Goto, Tomohiro Tanaka, Tsunenori Ouchida, Mika K. Kaneko, Yukinari Kato

**Affiliations:** 1Department of Molecular Pharmacology, Tohoku University Graduate School of Medicine, 2-1 Seiryo-machi, Aoba-ku, Sendai 980-8575, Japan; s1930550@s.tsukuba.ac.jp (N.G.); tomohiro.tanaka.b5@tohoku.ac.jp (T.T.); k.mika@med.tohoku.ac.jp (M.K.K.); 2Department of Antibody Drug Development, Tohoku University Graduate School of Medicine, 2-1 Seiryo-machi, Aoba-ku, Sendai 980-8575, Japan; tsunenori.ouchida.d5@tohoku.ac.jp

**Keywords:** CD44 variant 8, monoclonal antibody, gastric cancer, flow cytometry, immunohistochemistry

## Abstract

Gastric cancer (GC) is the third leading cause of cancer-related deaths worldwide. GC with peritoneal metastasis exhibits a poor prognosis due to the lack of effective therapy. A comprehensive analysis of malignant ascites identified the genomic alterations and significant amplifications of cancer driver genes, including *CD44*. CD44 and its splicing variants are overexpressed in tumors, and play crucial roles in the acquisition of invasiveness, stemness, and resistance to treatments. Therefore, the development of CD44-targeted monoclonal antibodies (mAbs) is important for GC diagnosis and therapy. In this study, we immunized mice with CD44v3–10-overexpressed PANC-1 cells and established several dozens of clones that produce anti-CD44v3–10 mAbs. One of the clones (C_44_Mab-94; IgG_1_, kappa) recognized the variant-8-encoded region and peptide, indicating that C_44_Mab-94 is a specific mAb for CD44v8. Furthermore, C_44_Mab-94 could recognize CHO/CD44v3–10 cells, oral squamous cell carcinoma cell line (HSC-3), or GC cell lines (MKN45 and NUGC-4) in flow cytometric analyses. C_44_Mab-94 could detect the exogenous CD44v3–10 and endogenous CD44v8 in western blotting and stained the formalin-fixed paraffin-embedded gastric cancer cells. These results indicate that C_44_Mab-94 is useful for detecting CD44v8 in a variety of experimental methods and is expected to become usefully applied to GC diagnosis and therapy.

## 1. Introduction

Gastric cancer (GC) is the third leading cause of cancer-related deaths globally [1]. The GC incidence is higher in Eastern Asia than in Western countries [2]. The vast majority of GC are adenocarcinomas, which can be divided into intestinal-type gastric cancer (IGC), diffuse-type gastric cancer (DGC), and mixed histology according to the Lauren classification [3]. The World Health Organization classifies gastric adenocarcinomas into papillary, tubular, mucinous, and poorly cohesive carcinomas [4]. Furthermore, next-generation sequencing defined four molecular subtypes, including Epstein–Barr virus-positive, microsatellite instability, genomically stable, and chromosomally unstable types [5,6]. The analysis also revealed the alterations in the GC genome and provided treatment options with anti-human epidermal growth factor receptor 2 (HER2) therapy [7] or immune checkpoint inhibitor therapy [8]. However, the benefit of those therapies is limited to a small subset of patients. In patients with advanced GC, especially those with DGC, peritoneal metastasis and subsequent development of malignant ascites are the most frequent cause of death [9]. Tanaka et al., therefore, performed a comprehensive multi-omic analysis of malignant ascitic samples and their corresponding tumor cell lines [10]. They identified the genomic alterations and significant amplification of known cancer driver genes, such as *KRAS*, *FGFR2*, *MET*, *ERBB2*, *EGFR, MYC*, *CCND1*, and *CD44* in GC with peritoneal metastasis [10]. Among them, the cell surface antigens (FGFR2, MET, HER2, EGFR, and CD44) are potentially treatable with monoclonal antibody (mAb) therapy [11]. In contrast to the first four antigens, neither therapy nor diagnosis against CD44 with mAb has yet been established.

CD44 plays important roles in tumor progression and has various isoforms [12], which are generated by the alternative splicing of CD44 mRNA [13]. The mRNA of the CD44 standard (CD44s) isoform is produced by constant region exons including the first five (referred to as exons 1 to 5) and the last five (16 to 20) [14]. The mRNAs of CD44 variant (CD44v) isoform are produced by the assembling of variant exons (v1–v10) with the constant region exons of CD44s [15]. CD44s and CD44v receive post-translational modifications, such as *N*-glycosylation and *O*-glycosylation [16]. Therefore, the molecular weight of CD44s is 80~100 kDa, while CD44v has various molecular weights (100~250 kDa) due to a variety of glycosylation [17]. Both CD44s and CD44v (pan-CD44) can attach to hyaluronic acid (HA), which is important for cellular adhesion, homing, and motility [18].

CD44v is overexpressed in tumors [12] and promotes tumor malignant progression through the binding to growth factors and the acquisition of invasiveness, stemness, and drug resistance [19,20,21]. These were mediated by the unique functions of the variant’s exon-encoded region. The v3-encoded region can recruit heparin-binding growth factors to their receptor and promote the signal transduction [22,23]. The v6-encoded region forms a ternary complex with hepatocyte growth factor and its receptor MET, which is essential for the activation [24,25]. Furthermore, the v8–10-encoded region binds to and stabilizes a cystine–glutamate transporter (xCT), which enhances cystine uptake and glutathione synthesis [26]. The elevation of reduced glutathione (GSH) mediates the defense against reactive oxygen species (ROS) [26] and chemotherapeutic drugs [27]. The expression of CD44v8–10 is associated with the function of xCT and intracellular redox status, which in turn is associated with poor prognosis [28]. Therefore, the establishment of CD44v-specific mAbs is essential for CD44-targeted tumor diagnosis and therapy. However, the roles played in tumor development by the variant 8-encoded region have not been fully elucidated.

Our group previously established an anti-pan-CD44 mAb—C_44_Mab-5 (IgG_1_, kappa) [29]—using the Cell-Based Immunization and Screening (CBIS) method. Moreover, another anti-pan-CD44 mAb—C_44_Mab-46 [30]—was developed by immunizing mice with CD44v3–10 ectodomain. C_44_Mab-5 and C_44_Mab-46 have epitopes within the standard exon 2 and 5-encoding regions, respectively [31,32]. We further showed that both C_44_Mab-5 and C_44_Mab-46 are applicable to flow cytometry and immunohistochemical analyses in oral squamous cell carcinomas (OSCC) [29] and esophageal SCC [30]. Furthermore, we produced a class-switched and a defucosylated version of C_44_Mab-5 (5-mG_2a_-f) using fucosyltransferase 8-deficient ExpiCHO-S cells and evaluated the antitumor effects of 5-mG_2a_-f in OSCC xenograft-bearing mice [33]. We have developed various anti-CD44v mAbs, including anti-CD44v4 (C_44_Mab-108) [34], anti-CD44v5 (C_44_Mab-3) [35], anti-CD44v6 (C_44_Mab-9) [36], anti-CD44v7/8 (C_44_Mab-34) [37], and anti-CD44v9 (C_44_Mab-1) [38].

In this study, we established a novel anti-CD44v8 mAb—C_44_Mab-94 (IgG_1_, kappa)—using the CBIS method and evaluated its applications.

## 2. Materials and Methods

### 2.1. Cell Lines

The human OSCC cell line (HSC-3) and the human gastric cancer cell lines (MKN45 and NUGC-4) were obtained from the Japanese Collection of Research Bioresources (Osaka, Japan). The human pancreatic cancer cell line (PANC-1) was obtained from the Cell Resource Center for Biomedical Research Institute of Development, Aging, and Cancer at Tohoku University (Sendai, Japan). Chinese hamster ovary (CHO)-K1 and P3X63Ag8U.1 (P3U1; a mouse multiple myeloma) cell lines were obtained from the American Type Culture Collection (ATCC, Manassas, VA, USA). HSC-3 was cultured in DMEM medium (Nacalai Tesque, Inc., Kyoto, Japan), supplemented with 100 μg/mL streptomycin, 100 U/mL penicillin, and 0.25 μg/mL amphotericin B (Nacalai Tesque, Inc.), and 10% (*v/v*) heat-inactivated fetal bovine serum (FBS; Thermo Fisher Scientific, Inc., Waltham, MA, USA). The cell lines (MKN45, NUGC-4, PANC-1, CHO-K1, and P3U1) were cultured in RPMI-1640 medium (Nacalai Tesque, Inc.), supplemented as indicated above. All cells were cultured using a humidified incubator at 37 °C, in an atmosphere of 5% CO_2_ and 95% air.

### 2.2. Construction of Plasmid DNA and Establishment of Stable Transfectants

The cDNAs of CD44s and CD44v3–10 were obtained as described previously [29]. The cDNAs were cloned into pCAG-zeo-ssPA16 and pCAG-neo-ssPA16 vectors with a signal sequence and N-terminal PA16 tag (GLEGGVAMPGAEDDVV). The PA16 tag can be detected by NZ-1 mAb, which was originally developed as an anti-human podoplanin (PDPN) mAb. N-terminal PA16-tagged CD44v3–10 deletion mutants (dN224, dN266, dN304, dN343, dN386, dN430, dN464, dN494, and dN562) were produced using a HotStar HiFidelity Polymerase Kit, and subcloned into the pCAG-neo-ssPA16 vector. The pCAG-zeo-ssPA16-CD44s, pCAG-zeo-ssPA16-CD44v3–10, and pCAG-neo/PA16-CD44v3–10 deletion mutant vectors were transfected into CHO-K1 cells. The pCAG-neo/PA16-CD44v3–10 vector was transfected into PANC-1 cells. The transfection was performed using a Neon transfection system (Thermo Fisher Scientific, Inc.). By the limiting dilution method, stable transfectants PANC-1/CD44v3–10, CHO/CD44s, CHO/CD44v3–10, and several deletion mutants of CHO/CD44v3–10 (dN224, dN266, dN304, dN343, dN386, dN430, dN464, dN494, and dN562) were finally established.

### 2.3. Production of Hybridomas

The 6-week-old female BALB/c mice were purchased from CLEA Japan (Tokyo, Japan). The mice were intraperitoneally immunized with PANC-1/CD44v3–10 (1 × 10^8^ cells) and Imject Alum (Thermo Fisher Scientific Inc.). Additional immunizations of PANC-1/CD44v3–10 (1 × 10^8^ cells, three times) and a booster injection of PANC-1/CD44v3–10 (1 × 10^8^ cells) was performed 2 days before the sacrifice. Hybridomas were produced as described previously [35]. The supernatants, which are positive for CHO/CD44v3–10 cells and negative for CHO-K1 cells, were selected by flow cytometry, SA3800 Cell Analyzers (Sony Corp., Tokyo, Japan).

### 2.4. Enzyme-Linked Immunosorbent Assay (ELISA)

Four peptides, covering the v7, v8, and v9 regions of CD44v3–10, were obtained from Sigma-Aldrich Corp. (St. Louis, MO, USA).

The peptide sequences were as follows.

CD44p421–440 (GHQAGRRMDMDSSHSTTLQP); v7/v8,

CD44p431–450 (DSSHSTTLQPTANPNTGLVE); v8,

CD44p441–460 (TANPNTGLVEDLDRTGPLSM); v8,

CD44p451–470 (DLDRTGPLSMTTQQSNSQSF); v8/v9.

The peptides (10 µg/mL) were immobilized on 96-well immunoplates (Nunc Maxisorp; Thermo Fisher Scientific Inc.). The immunoplate washing was performed with PBS containing 0.05% (*v/v*) Tween 20 (PBST; Nacalai Tesque, Inc.). The blocking was performed with 1% (*w/v*) bovine serum albumin (BSA) in PBST. C_44_Mab-94 (10 µg/mL) or blocking buffer was added to the peptide-coated wells. Next, the wells were further incubated with anti-mouse immunoglobulins conjugated with peroxidase (1:2000 dilution; Agilent Technologies Inc., Santa Clara, CA, USA). The ELISA POD Substrate TMB Kit (Nacalai Tesque, Inc.) was used for the peroxidase reaction. Using an iMark microplate reader (Bio-Rad Laboratories, Inc., Berkeley, CA, USA), the optical density (655 nm) was measured.

### 2.5. Flow Cytometry

HSC-3, MKN45, NUGC-4, CHO-K1, CHO/CD44s, CHO/CD44v3–10, and CHO/CD44v3–10 deletion mutants were obtained using 0.25% trypsin and 1 mM ethylenediamine tetraacetic acid (EDTA; Nacalai Tesque, Inc.). In the CBIS screening and epitope mapping, the hybridoma supernatants were treated with CHO-K1, CHO/CD44v3–10, or CHO/CD44v3–10 deletion mutants. In the dose-dependent assay, the cells were incubated with C_44_Mab-94, C_44_Mab-46, or control blocking buffer (0.1% BSA in PBS). Next, the cells were treated with anti-mouse IgG conjugated with Alexa Fluor 488 (1:2000; Cell Signaling Technology, Inc., Danvers, MA, USA). The data were analyzed using the EC800 Cell Analyzer and EC800 software ver. 1.3.6 (Sony Corp.), or the SA3800 Cell Analyzer and SA3800 software ver. 2.05 (Sony Corp.).

### 2.6. Western Blot Analysis

Cell lysates were prepared using NP-40 lysis buffer (20 mM Tris-HCl [pH 7.5], 150 mM NaCl, 1% Nonidet P-40, and 50 µg/mL of aprotinin) and were denatured in sodium dodecyl sulfate (SDS) sample buffer (Nacalai Tesque, Inc.). The 10 μg of proteins were subjected to electrophoresis using polyacrylamide gels (5–20%; FUJIFILM Wako Pure Chemical Corporation, Osaka, Japan) and transferred onto polyvinylidene difluoride (PVDF) membranes (Merck KGaA, Darmstadt, Germany). The membranes were blocked with 4% skim milk (Nacalai Tesque, Inc.) in PBST, and were incubated with 10 μg/mL of C_44_Mab-94, 10 μg/mL of C_44_Mab-46, or 0.5 μg/mL of an anti-β-actin mAb (clone AC-15; Sigma-Aldrich Corp.). Next, the membranes were incubated with anti-mouse immunoglobulins conjugated with peroxidase (diluted 1:1000; Agilent Technologies, Inc.). The chemiluminescence was performed with ImmunoStar LD (FUJIFILM Wako Pure Chemical Corporation) and detected using a Sayaca-Imager (DRC Co., Ltd., Tokyo, Japan).

### 2.7. Immunohistochemical Analysis

Formalin-fixed paraffin-embedded (FFPE) tissue arrays of gastric carcinoma (BS01012e and BS01011b) and OSCC (OR601c) were obtained from US Biomax Inc. (Rockville, MD, USA). The tissue arrays were autoclaved in citrate buffer (pH 6.0; Nichirei Biosciences, Inc., Tokyo, Japan) for 20 min. The blocking was performed using SuperBlock T20 (Thermo Fisher Scientific, Inc.). The sections were incubated with C_44_Mab-94 (5 μg/mL) and C_44_Mab-46 (5 μg/mL), and then treated with the EnVision+ Kit for mouse (Agilent Technologies Inc.). The chromogenic reaction was performed using 3,3′-diaminobenzidine tetrahydrochloride (DAB; Agilent Technologies Inc.). Hematoxylin (FUJIFILM Wako Pure Chemical Corporation) was used for counterstaining. A Leica DMD108 (Leica Microsystems GmbH, Wetzlar, Germany) was used to obtain images and examine the sections.

## 3. Results

### 3.1. Establishment of an Anti-CD44v8 mAb, C_44_Mab-94

We previously used CHO/CD44v3–10 cells as an immunogen and generated anti-CD44 mAbs, including C_44_Mab-5 (pan-CD44), C_44_Mab-3 (v5) [35], C_44_Mab-9 (v6) [36], and C_44_Mab-1 (v9) [38]. In this study, we established a new stable transfectant (PANC-1/CD44v3–10 cells) as another immunogen (Figure 1A). Mice were immunized with PANC-1/CD44v3–10 cells (Figure 1B), and hybridomas were seeded in 96-well plates (Figure 1C). The supernatants, which are positive for CHO/CD44v3–10 cells and negative for CHO-K1 cells, were selected using flow cytometry (Figure 1D). After cloning by limiting dilution, anti-CD44-mAb-producing clones were finally established. Among established clones, we focused on C_44_Mab-94 (IgG_1_, kappa), and the epitopes were determined by flow cytometry and/or ELISA (Figure 1E).

To determine the C_44_Mab-94 epitope, we examined the reactivity to CHO/CD44v3–10 and the N-terminal CD44v3–10 deletion mutants (dN224, dN266, dN304, dN343, dN386, dN430, dN464, dN494, and dN562)-expressed CHO-K1 cells by flow cytometry (Figure 2A). As shown in Figure 2B, C_44_Mab-94 reacted with dN224, dN266, dN304, dN343, dN386, dN430, and CD44v3–10. In contrast, the reactivity completely disappeared in dN464, dN494, and dN562. Because CD44v3–10 and the deletion mutants possess PA16 tag at the N-terminus, we could confirm all expression on the cell surface by an anti-PA16 tag mAb, NZ-1 (Figure 2C). These results suggest that C_44_Mab-94 recognizes the v8-encoding sequence.

To further assess the C_44_Mab-94 epitope, we performed ELISA using synthetic peptides from the v7- to v9-encoded sequences. As shown in Figure 3, C_44_Mab-94 reacted with CD44p431–450 (DSSHSTTLQPTANPNTGLVE, v8 region), but not with another v8 region (CD44p441–460), or v7/v8 region (CD44p421–440), or v8/v9 region (CD44p451–470). These results indicated that C_44_Mab-94 recognizes the CD44 variant-8-encoded sequence, but not the border sequence between v7 and v8, or between v8 and v9.

### 3.2. Flow Cytometric Analysis of C_44_Mab-94 against CD44-Expressing Cells

We next examined the reactivity of C_44_Mab-94 against CHO/CD44v3–10 and CHO/CD44s cells by flow cytometry. C_44_Mab-94 recognized CHO/CD44v3–10 cells in a dose-dependent manner (Figure 4A). In contrast, C_44_Mab-94 recognized neither CHO/CD44s (Figure 4B) nor CHO-K1 (Figure 4C) cells. We confirmed that a pan-CD44 mAb, C_44_Mab-46 [30], recognized both CHO/CD44s and CHO/CD44v3–10 cells (Appendix A, respectively), but not CHO-K1 cells (Appendix A). Furthermore, C_44_Mab-94 also recognized the OSCC cell line (HSC-3) and GC cell lines (MKN45, and NUGC-4) in a dose-dependent manner (Figure 4D–F, respectively).

### 3.3. Western Blot Analysis

We next conducted western blot analysis to assess the sensitivity of C_44_Mab-94. Total cell lysates from CHO-K1, CHO/CD44s, CHO/CD44v3–10, HSC-3, MKN45, and NUGC-4 were examined. As shown in Figure 5A, C_44_Mab-94 mainly detected CD44v3–10 at bands of ~75 kDa and over 180 kDa. Furthermore, C_44_Mab-94 detected endogenous CD44v8 at bands of over 100 kDa in HSC-3, MKN45, and NUGC-4 cells. An anti-pan-CD44 mAb, C_44_Mab-46, recognized the lysates from both CHO/CD44s (~75 kDa) and CHO/CD44v3–10 (>180 kDa) (Figure 5B). Although C_44_Mab-46 strongly recognized the lysates from NUGC-4, the reactivity to the HSC-3 and MKN45 lysates was weak. These results indicated that C_44_Mab-94 specifically detects exogenous CD44v3–10 and endogenous CD44v8.

### 3.4. Immunohistochemical Analysis Using C_44_Mab-94 against Tumor Tissues

We next investigated whether C_44_Mab-94 could be applied to immunohistochemical analysis using FFPE sections. We first examined the reactivity of C_44_Mab-94 in the OSCC tissue array because this type was revealed as the second highest CD44-positive cancer type in the Pan-Cancer Atlas [40]. As shown in Appendix A, the membranous staining in OSCC was observed by C_44_Mab-94 and C_44_Mab-46. In a stromal-invaded OSCC section, C_44_Mab-94 strongly stained invaded OSCC and could clearly distinguish tumor cells from stromal tissues (Appendix A). In contrast, C_44_Mab-46 stained both invaded OSCC and surrounding stroma cells Appendix A summarizes the results of OSCC tissue staining.

We next stained the GC tissue array (BS01011b) using C_44_Mab-94 and C_44_Mab-46. C_44_Mab-94 exhibited membranous staining in IGC (Figure 6A). C_44_Mab-46 also stained the same type of cancer cells (Figure 6B). Furthermore, membranous and cytoplasmic staining by C_44_Mab-94 and C_44_Mab-46 was observed in stromal-invaded tumor cells (Figure 6C,D). In DGC (Figure 6E,F), diffusely spread tumor cells were strongly stained by both C_44_Mab-94 and C_44_Mab-46. In contrast, neither C_44_Mab-94 nor C_44_Mab-46 stained the ductal epithelial structure of IGC (Figure 6G,H). Additionally, stromal staining by C_44_Mab-46 was observed in the tissue (Figure 6H).

We summarized the immunohistochemical analysis of GC in Table 1; C_44_Mab-94 stained 28 out of 72 cases (39%) of GC. A similar staining was also observed in another tissue array (BS01012e, Appendix A). We summarized the data of immunohistochemical analysis in Appendix A. These results indicated that C_44_Mab-94 is useful to detect CD44v8 in immunohistochemical analysis of FFPE tumor sections.

## 4. Discussion

The VFF series anti-CD44v mAbs were previously established by the immunization of bacterial-expressed CD44v3–10 and glutathione *S*-transferase fusion protein [41,42]. The clones, VFF-8 (v5), VFF-18 (v6), VFF-9 (v7), VFF-17 (v7/8), and VFF-14 (v10) have been used for various applications [43,44,45]. Furthermore, VFF-18 was humanized as BIWA-4 [46], and developed to bivatuzumab-mertansine, an antibody-drug conjugate (ADC), for clinical trials [47,48]. An anti-CD44v3 mAb (clone 3G5) [49] and an anti-CD44v9 mAb (clone RV3) [26] were also developed and widely used for research studies. However, a CD44v8-specific mAb had not been developed.

In this study, we developed a novel anti-CD44v8 mAb—C_44_Mab-94—using the CBIS method (Figure 1). We determined the epitope as a v8-encoded region using deletion mutants of CD44 (Figure 2), and synthetic peptides (Figure 3). We have established anti-CD44 mAbs using CHO/CD44v3–10 [29,35,36,38], PANC-1/CD44v3–10 (in this study), or purified CD44v3–10 ectodomain [30,37] as antigens. We listed them in our original “Antibody Bank” (see Appendix A). However, clones which recognize the v8-encoded region were rare, suggesting that the region has low antigenicity and/or locates the inside of CD44v3–10 protein. Although the affinity of C_44_Mab-94 is low against target cells, C_44_Mab-94 can be applied to various applications, including flow cytometry (Figure 4), western blotting (Figure 5), and immunohistochemistry (Figure 6).

We confirmed that C_44_Mab-94 recognizes a synthetic peptide of the v8-encoded region (DSSHSTTLQPTANPNTGLVE), but not border regions (v7/v8 and v8/v9) by ELISA (Figure 3). The epitope region possesses multiple confirmed and predicted *O*-glycosylation sites [50]. C_44_Mab-94 recognized a ~75-kDa band in the lysate of CHO/CD44v3–10 (Figure 5A), which is similar to the predicted molecular size from the amino acids of CD44v3–10. Therefore, C_44_Mab-94 could recognize CD44v3–10 regardless of the glycosylation. A detailed epitope analysis and an investigation of the influence of glycosylation on C_44_Mab-94 recognition are required in future studies.

In a GC cell line, the major transcripts of CD44v, including CD44v3, 8–10, CD44v6–10, CD44v8–10, and CD44v3, 8 were identified [39] (Figure 1A). C_44_Mab-94 can cover all products of the transcripts and detect the broad CD44v-expressing GC. Since CD44v8–10 plays critical roles in the regulation of ROS defense and GC progression [26], an anti-CD44v9 mAb (clone RV3) has so far mainly been used in immunohistochemistry. Several studies revealed that CD44v9 is a predictive marker for the recurrence of GC [51] and a biomarker for GC patient selection and efficacy of the xCT inhibitor, sulfasalazine [52]. Further investigations are required to reveal the relationship between CD44v8 expression and clinical factors using C_44_Mab-94. Additionally, C_44_Mab-94 recognized both IGC (Figure 6A) and DGC (Figure 6E) in immunohistochemistry. It would be worthwhile investigating whether CD44v8 is expressed in a specific molecular subtype of GC [6] in a future study.

A comprehensive analysis of malignant ascites identified the amplifications of cancer driver genes including *CD44* [10]. Although the expression pattern of CD44v is not identified, CD44v8 is thought to be an important target for mAb therapy due to the commonly included region in GC [39]. We have shown the antitumor activity using class-switched and defucosylated IgG_2a_ recombinant mAbs [33]. The defucosylated IgG_2a_ mAbs were produced by CHO-K1 lacking fucosyltransferases 8; they exhibited potent ADCC activity in vitro, and suppressed the growth of xenograft [53]. Therefore, the production of defucosylated C_44_Mab-94 is one of the strategies used to evaluate the antitumor effect on GC with peritoneal metastasis in the preclinical model.

Clinical applications of a humanized anti-CD44v6 mAb (BIWA-4) bivatuzumab-mertansine drug conjugate to solid tumors failed because of skin toxicities [47,48]. The accumulation of the mertansine drug was thought to be a cause of the toxicity [47,48]. Human acute myeloid leukemia (AML) cells also express high levels of CD44 mRNA due to the suppression of methylation of the CpG islands in the promoter [54]. Furthermore, higher expression of CD44v6 was observed in AML patients with *FLT3* or *DNMT3A* mutations. Therefore, a mutated version of BIWA-4—called BIWA-8—was engineered to develop chimeric antigen receptors (CARs) for AML. The CD44v6 CAR-T cells exhibited potent anti-leukemic effects [54,55] indicating that CD44v6 is a rational target of CAR-T therapy for AML harboring *FLT3* or *DNMT3A* mutations. Additionally, the CD44v6 CAR-T also showed an antitumor effect in lung and ovarian cancer xenograft models [56], which is expected for a wider development toward solid tumors.

Since CD44 mRNA is elevated in AML, other CD44 variants might also be transcribed in AML. Furthermore, CD44v8–10 was elevated during chronic myeloid leukemia (CML) progression from chronic phase to blast crisis in a humanized mouse model, which is required for the maintenance of stemness of CML [57]. Therefore, in future, we will investigate the reactivity of C_44_Mab-94 against hematopoietic malignancy. Further studies are required to investigate the selective expression of CD44v8 in leukemia cells, but not in hematopoietic stem cells, to certify its safety as a CAR-T antigen.

In this study, we used tumor cell-expressed CD44v3–10 as an immunogen. This strategy is important for the establishment of cancer-specific mAbs (CasMabs). We previously developed PDPN-targeting CasMabs [58] and podocalyxin-targeting CasMabs [59], which recognize cancer-type aberrant glycosylation of the targets [60]. Anti-PDPN-CasMabs are currently applied to CAR-T therapy in preclinical models [61]. For CasMab development, we should do further screening of our established anti-CD44 mAbs by comparing the reactivity against normal cells. Anti-CD44 CasMabs could be applicable for designing the modalities including ADCs and CAR-T.

## Figures and Tables

**Figure 1 antibodies-12-00045-f001:**
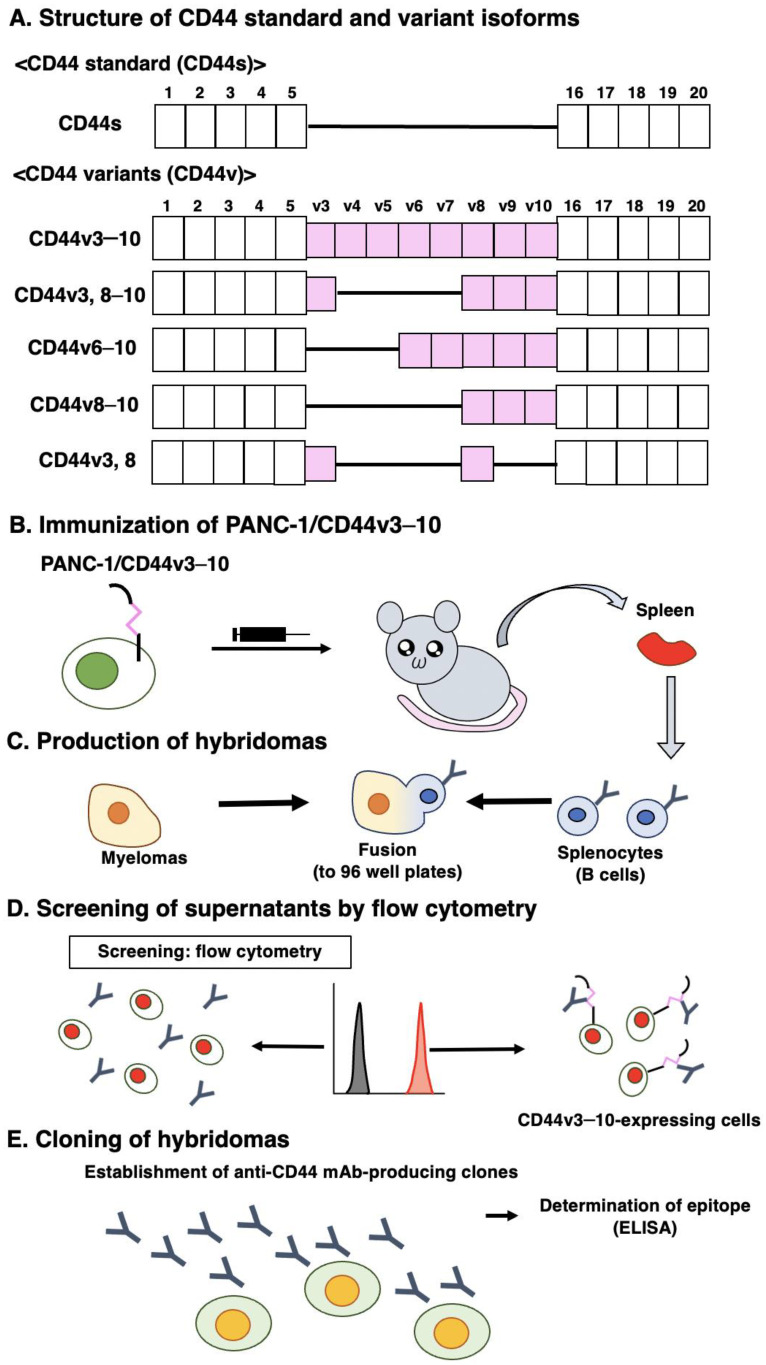
Anti-human CD44 mAbs production. (**A**) The structure of CD44s and CD44v. The mRNA of CD44s is assembled by the constant exons (1–5) and (16–20), and produces the standard isoform, CD44s. The mRNAs of CD44v are generated by the alternative splicing of variant exons. CD44v3–10 is an immunogen. CD44v3, 8–10, CD44v6–10, CD44v8–10, and CD44v3, 8 are detected in GC cell line [39]. (**B**) PANC-1/CD44v3–10 was used as an immunogen. (**C**) The hybridomas were produced by fusion with splenocytes and P3U1 cells. (**D**) The screening was performed using parental CHO-K1 and CHO/CD44v3–10 cells by flow cytometry. (**E**) A clone C_44_Mab-94 was established. Furthermore, the binding epitope was determined by flow cytometry using CD44 deletion mutant-expressed CHO-K1 cells and ELISA.

**Figure 2 antibodies-12-00045-f002:**
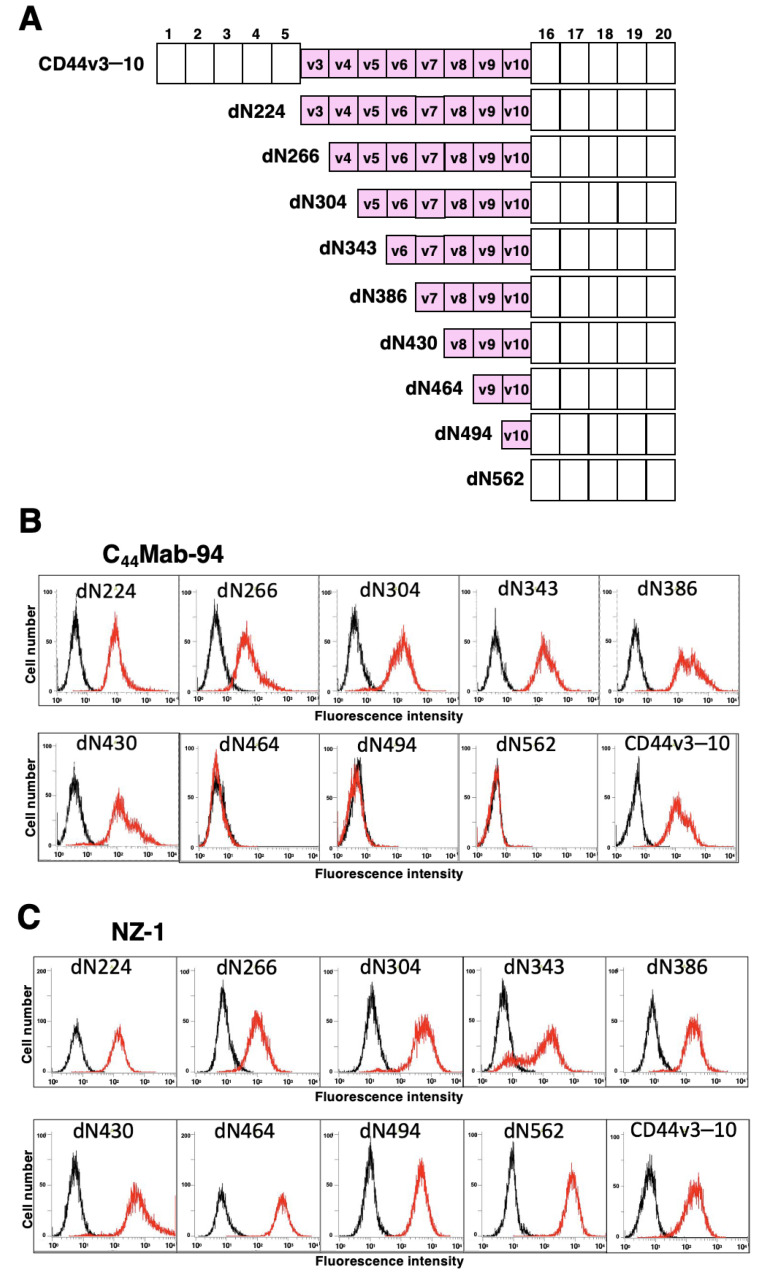
Epitope determination of C_44_Mab-94 using deletion mutants of CD44v3–10. (**A**) The CD44v3–10 deletion mutants expressed on CHO-K1 cells. (**B**) The CD44v3–10 mutants-expressed CHO-K1 cells were incubated with C_44_Mab-94 ((**B**), red line) or an anti-PA tag mAb, NZ-1 ((**C**), red line), or control blocking buffer (black line), followed by secondary antibodies treatment. The data were analyzed using the EC800 Cell Analyzer.

**Figure 3 antibodies-12-00045-f003:**
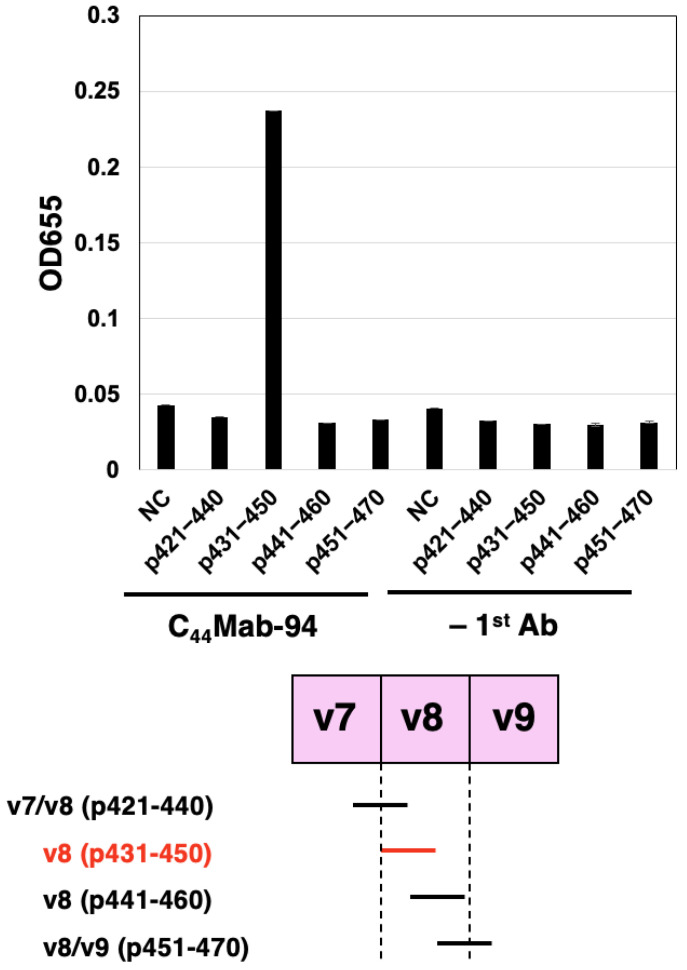
Determination of C_44_Mab-94 epitope by ELISA. Four synthesized peptides, which cover the CD44v7 to v9 region, were immobilized on immunoplates. The plates were incubated with C_44_Mab-94 or buffer control (–1st Ab), followed by incubation with peroxidase-conjugated anti-mouse immunoglobulins. Optical density was measured at 655 nm. ELISA—enzyme-linked immunosorbent assay. NC—negative control (solvent; DMSO in PBS). Error bars represent means ± SDs.

**Figure 4 antibodies-12-00045-f004:**
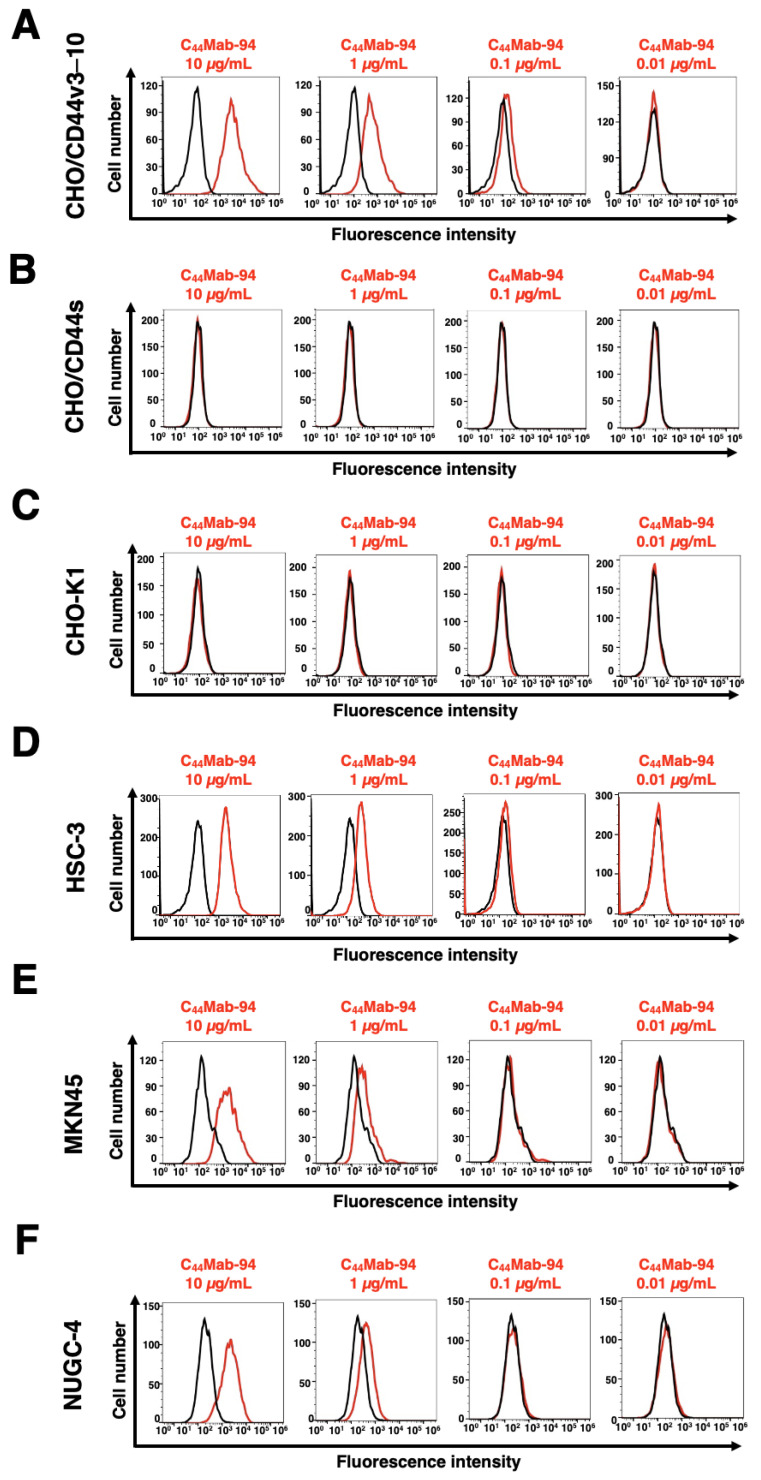
Flow cytometric analysis of C_44_Mab-94. CHO/CD44v3–10 (**A**), CHO/CD44s (**B**), CHO-K1 (**C**), HSC-3 (**D**), MKN45 (**E**), and NUGC-4 (**F**) cells were incubated with 0.01–10 µg/mL of C_44_Mab-94 (red line) or control blocking buffer (black line). Next, the cells were treated with anti-mouse IgG conjugated with Alexa Fluor 488. The data were analyzed using the SA3800 Cell Analyzer.

**Figure 5 antibodies-12-00045-f005:**
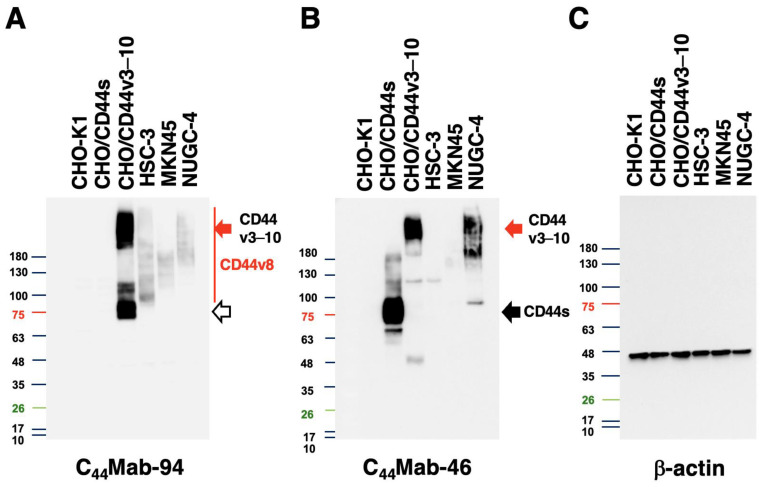
Western blot analysis using C_44_Mab-94. The cell lysates (10 µg) of CHO-K1, CHO/CD44s, CHO/CD44v3–10, HSC-3, MKN45, and NUGC-4 were electrophoresed and transferred onto polyvinylidene fluoride membranes. The membranes were incubated with 10 µg/mL of C_44_Mab-94 (**A**), 10 µg/mL of C_44_Mab-46 (**B**), and 0.5 µg/mL of an anti-β-actin mAb (**C**). Next, the membranes were incubated with anti-mouse immunoglobulins conjugated with peroxidase. The black arrows indicate CD44s (~75 kDa). The red arrows indicate CD44v3–10 (>180 kDa). CD44v8 was broadly detected in HSC-3, MKN45, and NUGC-4 lysates. The white arrow indicates a band at ~75 kDa recognized by C_44_Mab-94 in the CHO/CD44v3–10 lysate. A colored marker was used in the analysis.

**Figure 6 antibodies-12-00045-f006:**
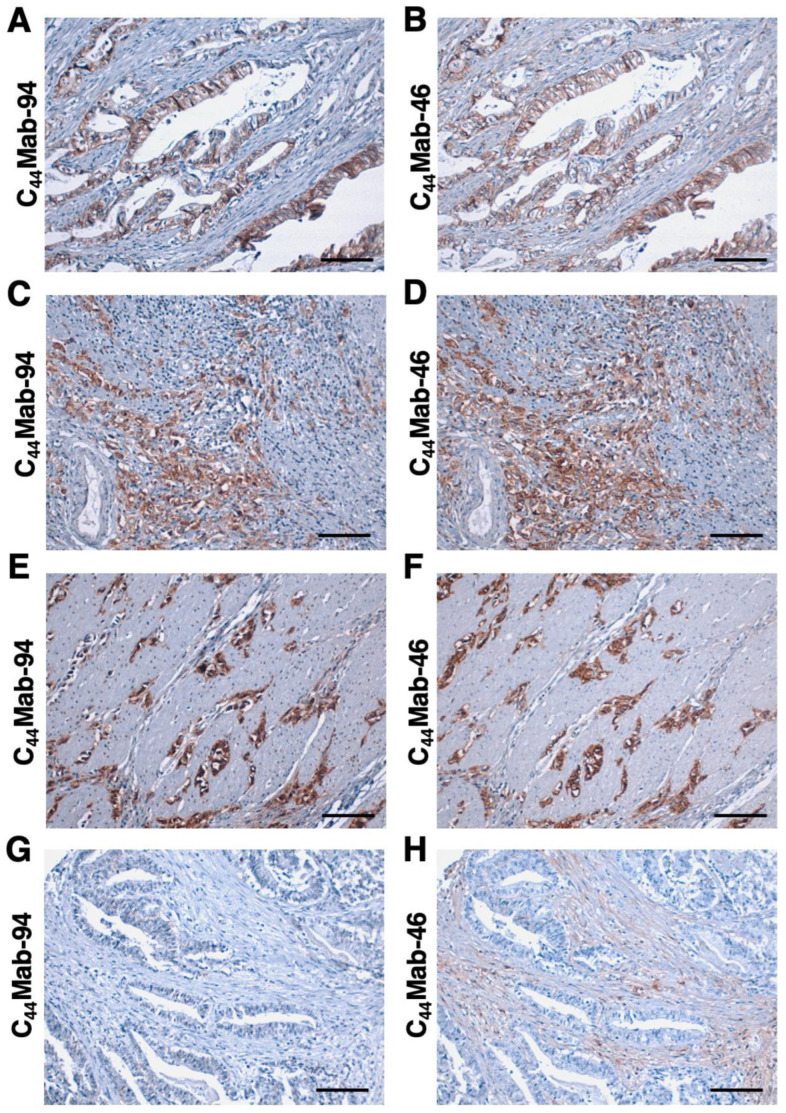
Immunohistochemistry using C_44_Mab-94 and C_44_Mab-46 against GC. (**A**–**H**) The serial sections of GC tissue arrays (BS01011b) were treated with 5 µg/mL of C_44_Mab-94 or 5 µg/mL of C_44_Mab-46, followed by treatment with the Envision+ kit. The chromogenic reaction was performed using DAB, and the sections were counterstained with hematoxylin. Scale bar = 100 µm.

**Table 1 antibodies-12-00045-t001:** Immunohistochemical analysis using C_44_Mab-94 and C_44_Mab-46 against GC tissue array (BS01011b).

No.	Age	Sex	Pathology Diagnosis	TNM	Grade	Stage	C_44_Mab-94	C_44_Mab-46
1	55	F	Adenocarcinoma	T2N0M0	1	IB	-	+
2	51	F	Adenocarcinoma	T2N0M0	-	IB	-	-
3	71	M	Adenocarcinoma	T3N1M0	1	IIB	-	++
4	63	M	Adenocarcinoma	T3N0M0	1	IIA	-	-
5	61	M	Adenocarcinoma	T2N0M0	1	IB	-	-
6	61	M	Adenocarcinoma	T2N0M0	1	IB	-	+
7	60	M	Adenocarcinoma	T3N2M0	1	IIIA	-	-
8	54	M	Adenocarcinoma	T3N2M0	1	IIIA	+	++
9	46	F	Adenocarcinoma	T3N0M0	1	IIA	-	+
10	66	M	Mucinous adenocarcinoma	T3N0M0	2–3	IIA	-	+
11	56	M	Adenocarcinoma	T2N0M0	2	IB	++	+
12	52	F	Adenocarcinoma	T3N0M0	2	IIA	+	+
13	70	M	Adenocarcinoma	T3N0M0	2	IIA	-	+
14	71	M	Adenocarcinoma	T2N0M0	2	IB	-	++
15	61	M	Adenocarcinoma	T3N0M0	2	IIA	-	-
16	75	M	Adenocarcinoma	T3N1M0	2	IIB	-	-
17	72	F	Adenocarcinoma	T3N0M0	2	IIA	+	+
18	60	M	Adenocarcinoma	T3N0M0	2	IIA	-	-
19	63	F	Adenocarcinoma	T3N0M0	2	IIA	-	-
20	69	M	Adenocarcinoma	T2N0M0	2	IB	-	-
21	54	F	Adenocarcinoma	T3N0M0	2	IIA	-	-
22	50	F	Adenocarcinoma	T3N0M0	3	IIA	-	-
23	64	M	Adenocarcinoma	T3N0M0	3	IIA	-	++
24	59	M	Adenocarcinoma	T2N0M0	2	IB	++	+++
25	59	M	Adenocarcinoma	T2N0M0	2	IB	-	-
26	44	M	Adenocarcinoma	T3N0M0	2	IIA	-	-
27	76	M	Adenocarcinoma	T3N0M0	2	IIA	-	+
28	56	M	Adenocarcinoma	T3N0M0	2	IIA	-	+
29	56	M	Adenocarcinoma	T2N0M0	2	IB	+	+
30	58	M	Adenocarcinoma	T3N0M0	2	IIA	-	+
31	94	M	Adenocarcinoma	T2N0M0	2	IB	+	+
32	56	F	Adenocarcinoma	T2N0M0	3	IB	+	+
33	56	M	Adenocarcinoma	T4N1M0	3	IIIA	+	+
34	51	F	Adenocarcinoma	T3N0M0	2	IIA	+	+
35	67	M	Adenocarcinoma	T3N0M0	2	IIA	+	+
36	53	M	Adenocarcinoma	T3N0M0	2–3	IIA	+	+
37	48	F	Adenocarcinoma	T2N1M0	3	IIA	++	++
38	58	M	Adenocarcinoma	T2N0M0	2	IB	-	-
39	61	M	Adenocarcinoma	T2N0M0	3	IB	-	+
40	62	M	Adenocarcinoma	T2N0M0	3	IB	-	+
41	65	M	Adenocarcinoma	T2N0M0	3	IB	+	+
42	47	F	Adenocarcinoma	T3N1M0	3	IIB	-	-
43	65	M	Adenocarcinoma	T2N0M0	-	IB	-	-
44	52	F	Adenocarcinoma	T2N0M0	3	IB	-	-
45	72	M	Adenocarcinoma	T2N0M0	3	IB	-	+
46	68	F	Adenocarcinoma	T3N0M0	3	IIA	-	++
47	56	M	Adenocarcinoma	T3N0M0	3	IIA	++	++
48	59	M	Adenocarcinoma	T3N1M0	3	IIB	+	+
49	62	M	Adenocarcinoma	T3N1M0	3	IIB	-	-
50	60	M	Adenocarcinoma	T3N1M0	3	IIB	+	+++
51	64	M	Adenocarcinoma	T2N0M0	3	IB	+	+
52	69	M	Adenocarcinoma	T2N0M0	3	IB	+	++
53	75	M	Adenocarcinoma	T2N0M0	3	IB	+	++
54	48	M	Adenocarcinoma	T2N0M0	3	IB	-	-
55	59	M	Adenocarcinoma	T2N0M0	3	IB	-	-
56	64	M	Adenocarcinoma	T3N0M0	3	IIA	+	+
57	55	M	Adenocarcinoma	T2N0M0	3	IB	++	++
58	58	M	Adenocarcinoma	T3N0M0	3	IIA	+	+
59	64	M	Adenocarcinoma	T3N0M0	3	IIA	+	+
60	67	M	Adenocarcinoma	T3N1M0	3	IIB	-	+
61	49	M	Adenocarcinoma	T2N0M0	3	IB	-	-
62	35	M	Adenocarcinoma	T3N1M0	3	IIB	-	-
63	45	F	Adenocarcinoma	T4N0M1	3	IV	+	++
64	43	M	Adenocarcinoma	T2N0M0	3	IB	-	+
65	56	M	Adenocarcinoma	T2N0M0	3	IB	-	-
66	66	M	Adenocarcinoma	T2N0M0	3	IB	+	+
67	60	M	Adenocarcinoma	T3N0M0	3	IIA	-	-
68	74	M	Adenocarcinoma	T2N0M0	3	IB	+	++
69	58	M	Adenocarcinoma	T2N0M0	3	IB	-	-
70	68	M	Mucinous adenocarcinoma	T2N0M0	2	IB	+	+
71	50	M	Mucinous adenocarcinoma	T3N0M0	3	IIA	-	-
72	51	M	Papillary adenocarcinoma	T2N0M0	2	IB	-	+

-, no stain; +, weak intensity; ++, moderate intensity; +++, strong intensity.

## Data Availability

The data presented in this study are available in the article and Appendix A.

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
