# Peer review of "Development of a Novel Anti-CD44 Variant 8 Monoclonal Antibody C44Mab-94 against Gastric Carcinomas"

_2073-4468, 2023, doi:10.3390/antib12030045_

Round 1

Reviewer 1 Report

Hiroyuki et al developed an anti-CD44 variant 8 Monoclonal antibodies designated as C44Mab-94. The antibody showed specificity for CD44 v8 and recognized the variant -8-encoded regions and peptides. The antibody has a dose-dependent binding manner with HSC-3 and GC cancer cell lines and can be applied by WB, Flow cytometry, and IHC. 

1. In Figure 4D-F, the Mab-94 showed a high binding with the cancer cell lines at 10ug/ml but Figure 5A showed basically unclear binding, the author should explain the reason.

2. Adding a positive Ctrl in the Flow cytometry and WB results would be better.

3. I suggest detecting the affinity difference between the conformational V8 region or CD44v3-10 and linear V8 or CD44v3-10 to address the different binding in Flow and WB.

4. How to calculate KD by Flow cytometry? Using MFI? The method is not precise, I suggest measuring the KD by BIAcore.

The English language can be minor revised 

Reviewer 2 Report

The manuscript antibodies-2414951 titled “Development of a Novel Anti-CD44 Variant 8 Monoclonal Antibody C44Mab-94 against Gastric Carcinomas”, focus on an interesting field, describing a development of a new specific anti-CD44v8 mAb C44Mab-94 for CD44-targeting tumor diagnosis and therapy, using the Cell-Based Immunization and Screening (CBIS).

The introduction provides sufficient background and includes all relevant and also recent references; the experimental design is clear and well written, and the results are well presented in the text.

In any way, some revisions are recommended.

I detected many self-citations by authors. This demonstrates that the present work is a logical continuation of the authors' previous works and demonstrates their expertise in the field, but the great innovative contribution of this presented study is not clear. There are more than 50% of references that are self-citations: 7,24-38; 40-47; 49; 51-53;66-72; 76-83.

Thus, a revision of the references is recommended.

Furthermore, I suggest changing the title font of reference n° 2 from uppercase to lowercase.

I suggest some other minor revisions:   

2.7 section- western blot analysis, page 4: specify briefly the preparation of cell lysates as described in ref 33 as reported in the 1st line   

in Results, figure 2 (B,C) page 6: the cytograms reported in figure 2 B and C would request the title on the X and Y axis   

in Results, Figure 3 page 7: graph lacks the title on the y-axis (Unit-measure)

Round 2

Reviewer 1 Report

The author claimed that Biocore is challenging to perform, instead, they chose Flow cytometry. But I believe the flow data could not indicate a KD value. They should change the description. 

Author Response

According to the reviewer's comment, we deleted the description of KD value in the result (3.2) and method sections.